# Regulatory T-cells inhibit microglia-induced pain hypersensitivity in female mice

Julia A Kuhn[1†‡], Ilia D Vainchtein[2†], Joao Braz[1], Katherine Hamel[1], Mollie Bernstein[1], Veronica Craik[1], Madelene W Dahlgren[3], Jorge Ortiz-Carpena[3], Ari B Molofsky[3], Anna V Molofsky[2]*, Allan I Basbaum[1]*

[1]Department of Anatomy, University of California San Francisco, San Francisco, United States; [2]Department of Psychiatry and Behavioral Sciences/Weill Institute for Neurosciences, University of California, San Francisco, San Francisco, United States; [3]Department of Laboratory Medicine, University of California, San Francisco, San Francisco, United States

**\*For correspondence:**
Anna.Molofsky@ucsf.edu (AVM);
allan.basbaum@ucsf.edu (AIB)

[†]These authors contributed equally to this work

**Present address:** [‡]Genentech, South San Francisco, United States

**Abstract** Peripheral nerve injury-induced neuropathic pain is a chronic and debilitating condition characterized by mechanical hypersensitivity. We previously identified microglial activation via release of colony-stimulating factor 1 (CSF1) from injured sensory neurons as a mechanism contributing to nerve injury-induced pain. Here, we show that intrathecal administration of CSF1, even in the absence of injury, is sufficient to induce pain behavior, but only in male mice. Transcriptional profiling and morphologic analyses after intrathecal CSF1 showed robust immune activation in male but not female microglia. CSF1 also induced marked expansion of lymphocytes within the spinal cord meninges, with preferential expansion of regulatory T-cells (Tregs) in female mice. Consistent with the hypothesis that Tregs actively suppress microglial activation in females, Treg deficient (*Foxp3*[DTR]) female mice showed increased CSF1-induced microglial activation and pain hypersensitivity equivalent to males. We conclude that sexual dimorphism in the contribution of microglia to pain results from Treg-mediated suppression of microglial activation and pain hypersensitivity in female mice.

## Introduction

Microglia are brain resident macrophages with essential roles in neural circuit function in physiology and disease (*Priller and Prinz, 2019*; *Hammond et al., 2018*; *Vainchtein and Molofsky, 2020*). Microglia respond in sexually dimorphic ways in a variety of contexts, including autism, stroke, neurodegenerative diseases, and interestingly in the microglial contribution to pain processing (*Mogil, 2020*; *Villa et al., 2018*; *Weinhard et al., 2018*; *Sorge et al., 2011*; *Inyang et al., 2019*; *Rosen et al., 2019*; *Kodama and Gan, 2019*; *Guneykaya et al., 2018*). For example, although male and female microglia are competent to induce pain (*Yi et al., 2021*), pharmacologic ablation or chemogenetic inhibition of microglia reverses peripheral nerve injury-induced mechanical hypersensitivity only in male mice (*Sorge et al., 2015*; *Saika et al., 2020*). In contrast, inhibition of microglia is sufficient to reverse injury-induced hypersensitivity in B- and T-cell deficient female mice (*Sorge et al., 2015*). Taken together, these data imply that there are sex-specific differences in how the innate and adaptive immune compartments interact to regulate neuropathic pain.

We previously identified microglial activation via release of the myeloid survival factor, colony-stimulating factor 1 (CSF1), from injured sensory neurons as a mechanism contributing to nerve injury-induced pain (*Guan et al., 2016*). Here, we show that intrathecal administration of CSF1 is sufficient

to induce pain (mechanical hypersensitivity) in male, but not female mice. Transcriptomic profiling of dorsal horn microglia and morphologic analyses demonstrated that this sex-specific effect correlates with robust microglial activation in male but not female mice. Furthermore, intrathecal CSF1 markedly expanded lymphocytes and myeloid cells in the spinal cord meninges, and resulted in a preferential expansion of regulatory T-cells (Tregs), in female mice. Finally, we demonstrate that Treg depletion (*FoxP3^DTR*) in female mice promotes CSF1-induced microglial activation and is sufficient to induce CSF1-induced pain hypersensitivity equivalent to males. Our findings reveal novel cross-regulatory interactions between Tregs and spinal cord microglia that modulate a sex-specific pain phenotype.

## Results

CSF1 is de novo expressed in injured sensory neurons (*Guan et al., 2016*), and in the spinal cord, parenchymal microglia are the only cells expressing CSF1 receptor (CSF1R). We first analyzed injury-induced mechanical hypersensitivity in female *Avil^Cre:Csf1^fl/fl* mice (Adv-CSF1) in which CSF1 is specifically deleted from sensory neurons. We found that female Adv-CSF1 mice developed normal mechanical hypersensitivity after peripheral nerve injury (*Figure 1—figure supplement Figure 1— figure supplement 1A, B*), in contrast to male rats and mice, in which hypersensitivity was CSF1-dependent (*Guan et al., 2016*; *Okubo et al., 2016*). Thus, CSF1 is not required to induce mechanical hypersensitivity in females.

We next assessed whether selective administration of CSF1, via an intrathecal route, is sufficient to induce mechanical hypersensitivity. Three consecutive injections of CSF1 provoked profound mechanical hypersensitivity in male, but not in female mice (*Figure 1A–C*), even at very high doses (30 ng; *Figure 1—figure supplement 1C*). Furthermore, after intrathecal CSF1, male microglia acquired a robust amoeboid morphology, characterized by loss of ramification, but in females, microglia acquired a highly ramified morphology, consistent with a persistent homeostatic phenotype (*Figure 1D–E*). Fluorescence-activated cell sorting (FACS) analysis also revealed larger numbers of microglia in males and higher expression of cell surface activation markers, CD11b/CD45 (*Figure 1F–H*, *Figure 1—figure supplement 1D*). Taken together, these data demonstrate a male-specific impact and sufficiency of CSF1 for microglia activation and pain hypersensitivity.

To determine whether there was a differential impact of CSF1 on male versus female microglia, we transcriptionally profiled flow-sorted microglia from the lumbar dorsal horn. Sex differences were modest at baseline (86 genes, $p_{Adj}$ <0.01), and CSF1 induced robust gene expression changes in both male and female microglia (*Figure 2A*, PC1, 56% of variance). However, CSF1 induced an 8.3-fold increase in differentially expressed genes (both upregulated and downregulated) in male microglia (*Figure 2B*, *Supplementary file 1*; adjusted p-value<0.01: males 1350 genes, females 165 genes). As CSF1 is an essential survival factor for microglia and myeloid cells, these sex-specific microglia responses to CSF1 were surprising. Neither the protein nor transcriptomic CSF1R levels differed between males and females (*Figure 2—figure supplement 1A, B*).

We next examined these gene expression changes in the context of published microglial transcriptomic data sets in homeostasis and disease (*Friedman et al., 2018*; *Figure 2C*). Both male and female microglia responded to CSF1 with a decrease in homeostatic gene expression and an increase in proliferative genes, which were more prominent in males than females. Most prominent in male microglia was a striking upregulation of pathology-associated microglial activation genes (*Figure 2C*; Neurodegeneration module) (*Friedman et al., 2018*; *Keren-Shaul et al., 2017*). Gene ontology (GO) enrichment analysis (*Figure 2D*) revealed that male microglia induced genes and GO terms that are linked to classical immune activation and recruitment pathways, including many (*Itgax, Lpl, Ccl3, Cybb, Clec7a*, and *Ctsb*) associated with the 'disease associated microglia' DAM phenotype identified in single-cell sequencing experiments (*Butovsky and Weiner, 2018*). Some of these genes, for example, *Ctsb*, have been linked to chronic pain (*Sun et al., 2012*). In addition, male microglia downregulated genes facilitating responsiveness to extracellular signals as well as some supportive functions, for example, extracellular matrix regulation (*Figure 2D*). Taken together, intrathecal CSF1 not only triggers pain hypersensitivity in male mice, but also induces robust transcriptomic changes associated with inflammatory activation in male but not female microglia.

Our findings suggest that other immune cells contribute to amplify or suppress the microglial response to CSF1. The CNS meninges have a rich population of immune cells that mirrors the composition of tissue resident immune cells in other organs (*Alves de Lima et al., 2020*; *Figure 3B*).

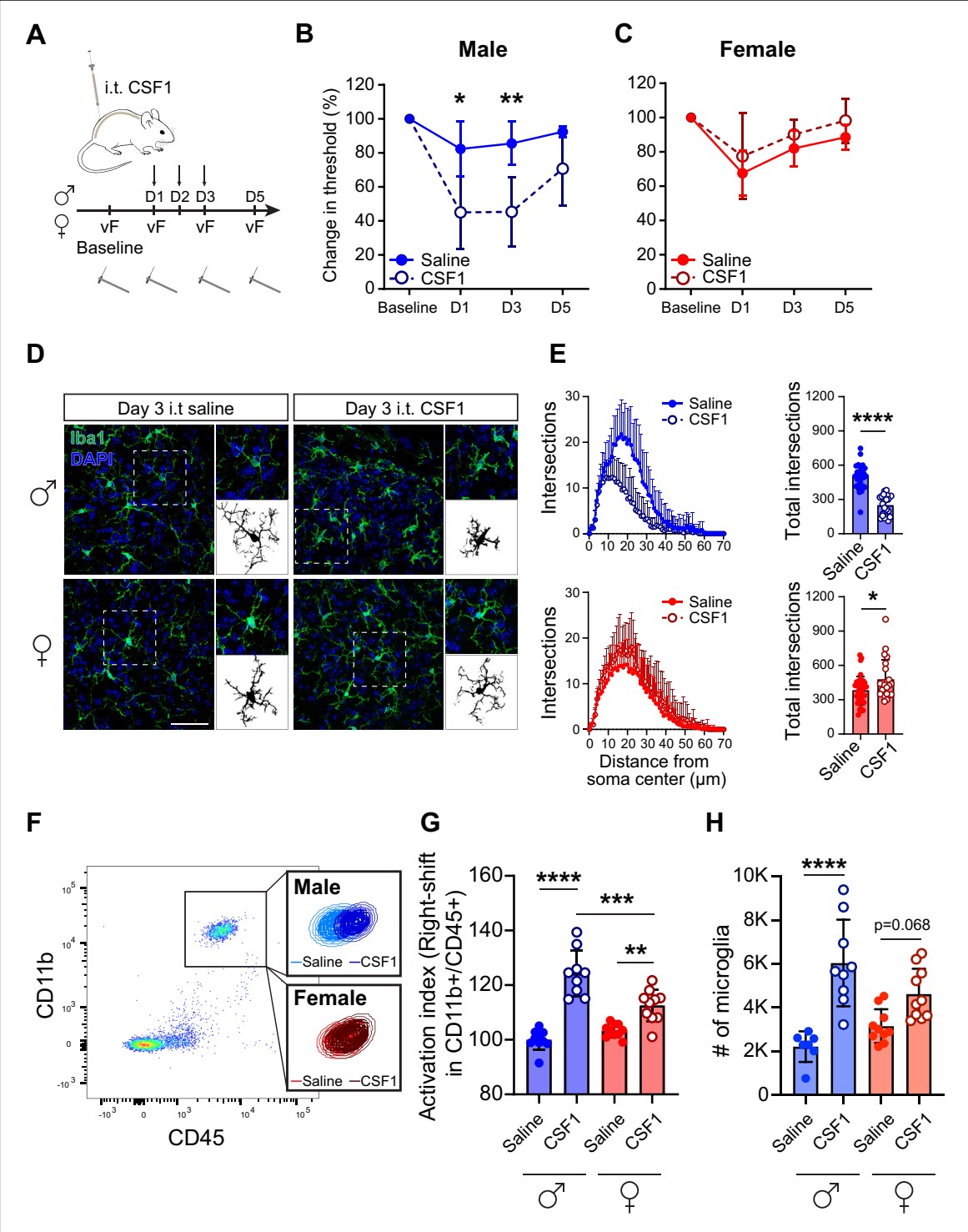

**Figure 1.** CSF1 induces pain hypersensitivity and microglial activation in male but not female mice. (**A**) Schematic depicting 3 days of CSF1 intrathecal injection (i.t.) paradigm with von Frey assay. (**B, C**) Change in mechanical pain threshold in males and females after saline or CSF1 injection. N=5–7 mice per condition, repeated measures ANOVA. (**D**) Representative immunohistochemistry of lumbar spinal cord sections after 3 days of CSF1 i.t. injection. Insets indicate single microglia and binary images used for subsequent quantifications. Scale bar=50 μm. (**E**) Ramification calculated by Scholl analysis in males (blue, top) and females (red, bottom). N=3 mice/condition, 25 cells/group; dots represent individual microglia, Student's t-test. (**F**) Representative flow cytometry plot demonstrating right-shift of the CD11b+/CD45+ population in lumbar spinal cord. Insets indicate microglia population gated on CD11b+CD45+Ly-6C−. (**G**) Microglial activation index calculated from flow-cytometry data as a sum of mean fluorescence intensity of CD11b and CD45 fluorescence intensity. Dots represent individual mice. One-way ANOVA with Tukey's multiple comparisons. (**H**) Microglial numbers calculated by flow

*Figure 1 continued on next page*

*Figure 1 continued*

cytometry data. Dots represent individual mice. One-way ANOVA with Tukey's multiple comparisons. *p<0.05, **p<0.01, ***p<0.001, ****p<0.0001. CSF1, colony-stimulating factor 1.

The online version of this article includes the following figure supplement(s) for figure 1:

**Figure supplement 1.** CSF1 deletion in sensory neurons rescues pain in male but not female mice.

Meningeal lymphocyte-derived cytokines also impact CNS function in both normal and pathologic settings (*Liu et al., 2020*; *Pasciuto et al., 2020*; *Ribeiro et al., 2019*). We examined the immune cell composition of spinal cord meninges using 11-parameter flow cytometry of dissociated meninges (*Figure 3—figure supplement 1A-C*, *Figure 3A–C*). As expected, intrathecal CSF1 expanded meningeal macrophages (*Figure 3—figure supplement 1B*), but we also observed a marked increase in lymphocytes, 6.5-fold in males and 9-fold in females (*Figure 3—figure supplement 1C*). Further examination of lymphocyte subsets demonstrated a similar increase of CD4+ FoxP3 T cells, CD8+ T cells, B cells, and ILC2 cells in male and female meninges, but also revealed a significantly greater expansion of regulatory T cells and natural killer (NK) cells in female mice (*Figure 3B–C*).

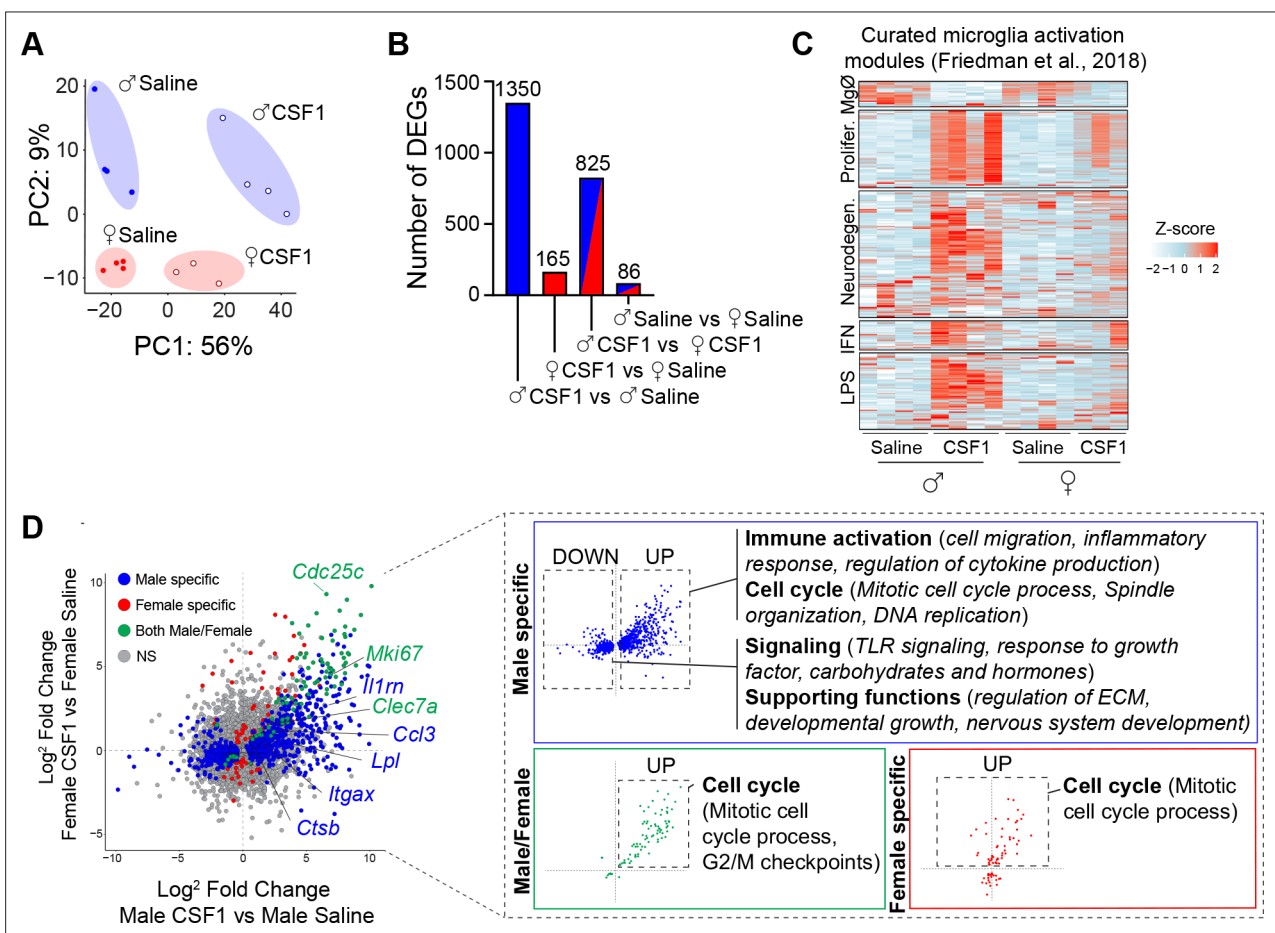

**Figure 2.** CSF1 promotes immune activation in male but not female microglia. (**A**) Principal component analysis of genes expressed by microglia isolated by flow cytometry from male and female mice after 3 days of saline or CSF1 i.t. Dots represent individual mice. (**B**) Number of differentially expressed genes (DEGs) per comparison (adjusted p-value<0.01). (**C**) Heatmap of DEGs in male and female microglia after CSF1 overlaid with microglia activation modules curated by *Friedman et al., 2018*. (**D**) Four-way plot depicting DEGs (adjusted p-values<0.01) that are male-specific (blue), female-specific (red), or male-female shared (green). Inset highlights gene ontology terms identified in the respective categories. CSF1, colony-stimulating factor 1; i.t., intrathecal injection.

The online version of this article includes the following figure supplement(s) for figure 2:

**Figure supplement 1.** Male and female microglia express equal levels of the CSF1 receptor.

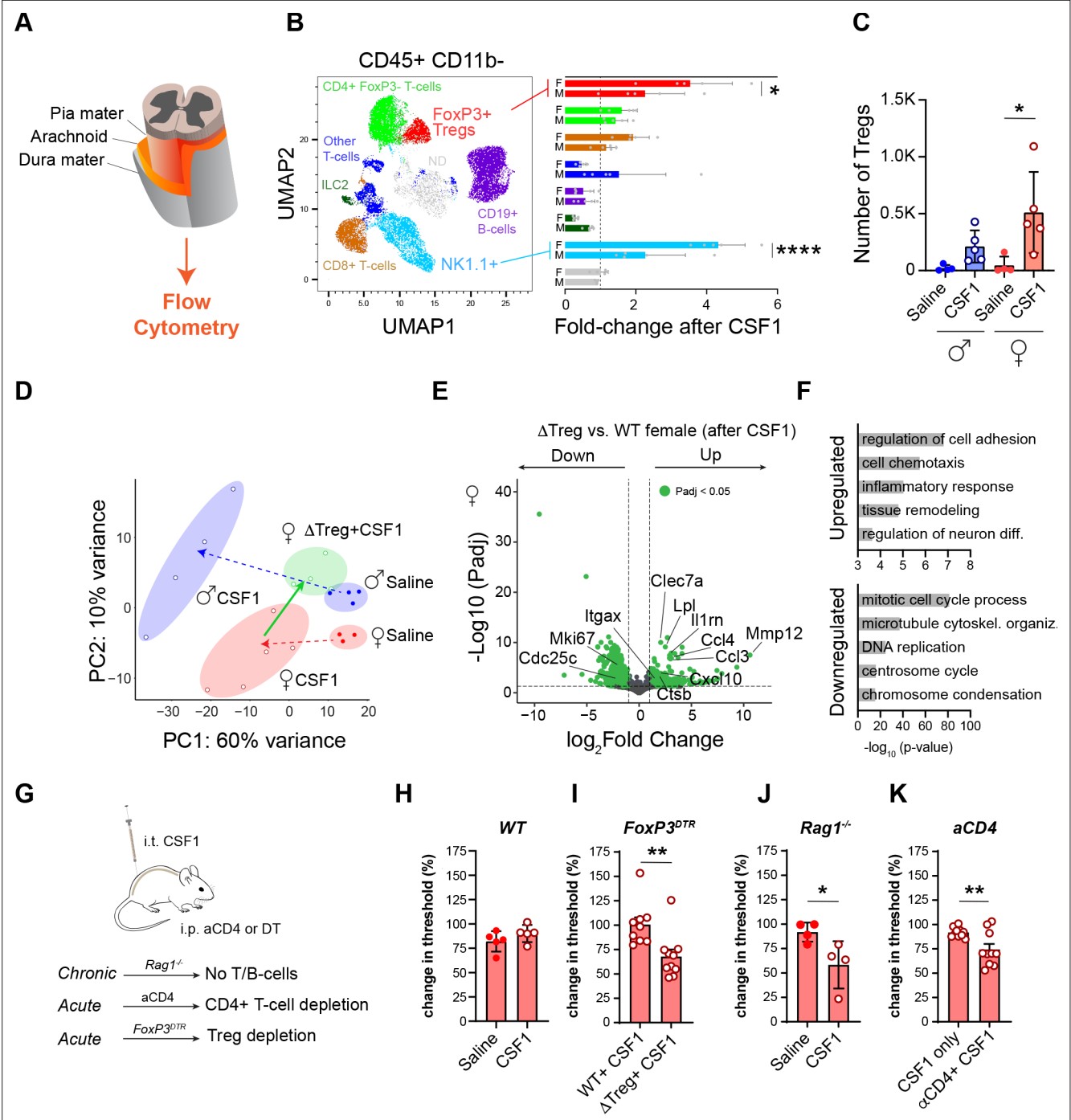

**Figure 3.** Regulatory T-cells restrict microglial activation and pain behavior in female mice. (**A**) Schematic of spinal cord meninges. (**B**) UMAP plot of lymphoid, non-myeloid cells (CD45$^+$CD11b$^-$) isolated from spinal cord meninges. Image is a pool of all samples colored by cell type specific markers as indicated. Bar graph shows fold-change in indicated populations in males and females after CSF1. Dots in bar graph: individual samples. N=5 mice per group. (**C**) Quantification of regulatory T-cells (Tregs; CD4$^+$FoxP3$^+$) from (**B**). (**D**) Principal component analysis (PCA) of microglial gene expression profiles in select conditions. Red=female, blue=male, green=Treg deficient female (*FoxP3$^{DTR}$*). Dots: individual mice. PCA consists of two experiments. The first experiment is depicted in ***Figure 2A*** and complemented with a second experiment consisting of WT females with CSF1 and Treg deficient females treated with CSF1. (**E**) Volcano plot depicting DEGs (adjusted p-values<0.05; green) between female Treg KO mice after CSF1 versus female mice after CSF1. N=4 mice per group. (**F**) Gene ontology terms for upregulated and downregulated genes from volcano plot in (**E**). (**G**) Schematic depicting the approach of using Rag1 KO mice (no T/B cells), antibody against CD4 (aCD4) to deplete T-cells and *FoxP3$^{DTR}$* mice, in which Tregs are depleted using diphtheria toxin. (**H, I**) Change in mechanical hypersensitivity at day 3 after i.t. CSF1 in WT female mice (data from day 3, ***Figure 1B***) or in females lacking regulatory T-cells (*FoxP3$^{DTR}$*). Dots: individual mice. (**J**) Change in mechanical hypersensitivity at day three after CSF1 i.t. in *Rag1$^{-/-}$*. Dots:

*Figure 3 continued on next page*

*Figure 3 continued*

individual mice. (**K**) Change in mechanical hypersensitivity at day 3 after CSF1 in female mice injected with a CD4 blocking antibody 1 day prior to CSF1 injections. Dots: individual mice. In (**I–K**) unpaired two-tailed t-test and (**C**) one-way ANOVA with Tukey's multiple testing correction. *p<0.05, **p<0.01, ****p<0.0001. DEG, differentially expressed gene; WT, wild-type.

The online version of this article includes the following figure supplement(s) for figure 3:

**Figure supplement 1.** Isolation and depletion of meningeal immune cells.

**Figure supplement 2.** T-cells are rarely detected 7 days post SNI.

As NK cells are traditionally considered pro-inflammatory including in the context of pain (*Greisen et al., 1999*; *Das et al., 2018*) and microglial activation (*Garofalo et al., 2020*), whereas Tregs are potent suppressors of inflammation, we next asked whether Tregs in females counter the CSF1-induced microglial activation and pain. To acutely deplete Tregs, we administered diphtheria toxin to *FoxP3^{DTR}* mice (*Sakaguchi et al., 2008*; *Ali et al., 2017*; *Da Costa et al., 2019*; *Kim et al., 2007*; *Figure 3—figure supplement 1D,E*). From these mice, we transcriptionally profiled female microglia after CSF1 intrathecal injection in the control or Treg depleted setting (*Figure 3D–F/Supplementary file 3*). We found that female microglia expressed many of the male-specific CSF1 induced genes, including genes involved in immune activation and recruitment (*Clec7a, Il1rn, Ccl3, Ccl4*, and *Ctsb*; *Figure 3E–F*). We also observed alterations of genes that are unique to the Treg-depleted context (*Figure 3—figure supplement 1F*). We conclude that Treg depletion partly restores the pro-inflammatory microglial response to CSF1 in female mice.

Finally, we tested whether Tregs suppress CSF1-induced mechanical hypersensitivity in female mice. We depleted Tregs in *FoxP3^{DTR}* mice by administering diphtheria toxin prior to CSF1 injection (*Figure 3G*). Compared to wild-type (WT) females, Treg depletion in females led to a 33% increase in mechanical hypersensitivity (*Figure 3H–I*; summarizes D3 timepoint from *Figure 1A*). This effect was phenocopied in *Rag1^{−/−}*, which lack T- and B-cells from birth but retain innate lymphocytes, such as NK cells (*Figure 3J*) and the findings are reminiscent of those reported in *Rag1^{−/−}* female after peripheral nerve injury (*Sorge et al., 2015*). Of note, depleting Tregs in males did not alter their mechanical hypersensitivity (*Figure 3—figure supplement 1G*). Acute antibody blockade of CD4^+ T-cells, which include both suppressive (Tregs) and inflammatory subsets (Th1/Th2), also phenocopied this increase in mechanical hypersensitivity (*Figure 3K*; *Figure 3—figure supplement 1H-I*). Taken together, we demonstrate that this difference reflects a suppressive effect of Tregs on the CSF1-mediated immune activation in female mice, rather than a direct pain-mediating effect of T-cells on dorsal horn pain circuitry.

## Discussion

Our identification of a sex-specific interaction between spinal cord microglia and Tregs that mediates male/female differences in a model of neuropathic pain has several important implications. First, we defined the immune activation profile of CSF1 on microglia in vivo and demonstrated robust expansion of lymphocytes within the spinal cord meninges in response to CSF1. These results are consistent with a model in which one function of CSF1-stimulated myeloid cells is to recruit other immune cells that in turn release cytokines and chemokines to impact microglial function. However, the nature of this immune response is strikingly sex-specific. In males, the balance tips toward pro-inflammatory signaling. In females, Tregs suppress inflammatory activation and limit mechanical hypersensitivity development, despite expansion of the myeloid and lymphoid compartments. As intrathecal CSF1 induces mechanical hypersensitivity in Treg-depleted female mice, we concur that female microglia are indeed competent to contribute to pain hypersensitivity (*Yi et al., 2021*; *Sorge et al., 2015*). However, our results demonstrate that CSF1-mediated cross-talk between spinal cord microglia and lymphocytes can either amplify or suppress pain phenotypes.

Our findings also introduce spinal cord meninges as a potentially relevant source of immune cells that coordinate microglial responses in the setting of neuropathic pain. Importantly, in contrast with a previous report (*Costigan et al., 2009*), we rarely detected lymphocytes, including T-cells, in the spinal cord, even after nerve injury (*Figure 3—figure supplement 2*). However, we found that immune cells markedly expand within the spinal cord meninges, even when absent from the parenchyma. As lymphocytes act primarily via secreted cytokines, we suggest that release of meningeal-derived

cytokines impacts microglial function as well as directly impacts nociceptors (*Liu et al., 2014*). Although our report focuses on the contribution of Tregs, we also detected a female-specific increase in meningeal NK cells in response to CSF1. NK cells are classically associated with pro-inflammatory responses, however, recent studies highlight their more diverse functions. These include instruction of anti-inflammatory astrocytes from meningeal NK cells (*Sanmarco et al., 2021*), beneficial effects after peripheral nerve injury (*Davies et al., 2019*), and a negative correlation between NK cells in the cerebrospinal fluid and mechanical pain sensitivity in chronic neuropathic pain patients (*Lassen et al., 2021*). The function of meningeal NK cells in CSF1-induced pain in mice remains to be determined.

In the setting of injury, inflammatory signaling at multiple access points (e.g., injury site, nerve, and DRG) activates nociceptive circuits (*Yu et al., 2020*). However, our finding that intrathecal activation of myeloid cells is sufficient to activate meningeal immunity raises the possibility that modulating the meninges is a potential therapeutic avenue of neuropathic pain management, by suppressing meningeal Treg expansion-mediated microglial activation or by the release of intrathecal immune modulators that override peripheral inflammatory cues. Given that human genetic analyses and other studies indicate a contribution of Tregs and their dominant cytokines in neuropathic and inflammatory pain models (*Davoli-Ferreira et al., 2020*; *Fischer et al., 2019*; *Milligan et al., 2006*; *Eijkelkamp et al., 2016*; *Echeverry et al., 2009*; *Kringel et al., 2018*), further investigations of Treg localization and impact on microglia will be relevant to understanding the generation and conceivably the treatment of nerve-injury-induced chronic pain.

# Materials and methods

**Key resources table**

| Reagent type (species) or resource | Designation | Source or reference | Identifiers | Additional information |
|---|---|---|---|---|
| Gene (*Mus musculus*) | *Csf1* | MGI | MGI:1339753 NCBI Gene: 12,977 | |
| Gene (*M. musculus*) | *Foxp3* | MGI | MGI:1891436 NCBI Gene: 20,371 | |
| Gene (*M. musculus*) | *Avil* | MGI | MGI:1333798 NCBI Gene: 11,567 | |
| Strain, strain background (*M. musculus, male and female*) | C57BL/6 J | The Jackson Laboratory | RRID:IMSR_JAX:000664 | |
| Strain, strain background (*M. musculus, male and female*) | B6.129S7-Rag1tm1Mom/J | The Jackson Laboratory | RRID:IMSR_JAX:002216 | |
| Strain, strain background (*M. musculus, male and female*) | B6.129(Cg)-*Foxp3*tm3(DTR/GFP)Ayr/J | The Jackson Laboratory | RRID:IMSR_JAX:016958 | |
| Strain, strain background (*M. musculus, male and female*) | *Avil*Cre | *Zurborg et al., 2011* | | |
| Strain, strain background (*M. musculus, male and female*) | *Csf1*fl/fl | *Harris et al., 2012* | | |
| Peptide, recombinant protein | CSF1 (*M. musculus*) | Thermo Fisher Scientific | Cat: #PMC2044 | 15 ng or 30 ng in 5 µl (i.t.) |
| Peptide, recombinant protein | Diphtheria Toxin (*Corynebacterium diphtheriae*) | Sigma-Aldrich | Cat: #D0564 | 30 ng/g in 200 µl (i.p.) |

*Continued*

| Reagent type (species) or resource | Designation | Source or reference | Identifiers | Additional information |
|---|---|---|---|---|
| Antibody | Monoclonal rat anti-mouse CD4 Clone: GK1.5 | Bio X Cell | Cat: #BE0003-1 | 250 µg in 200 µl (i.p.) |
| Antibody | Polyclonal Rabbit anti-mouse Iba1 | WAKO | Cat: #019-19741 | IF: (1:2000) |
| Antibody | Monoclonal Alexa 647-coupled rat anti-mouse CD45 (clone 30-F11) | BioLegend | Cat: #103123 | IF: (1:200) |
| Antibody | Monoclonal hamster anti-mouse CD3 (clone 145-2C11 ) | BD Bioscience | Cat: #553058 | IF: (1:200) |
| Antibody | Monoclonal PE anti-mouse CD11b (clone M01/70) | eBioscience | Cat: #12-0112-81 | FACS (1:200) |
| Antibody | Monoclonal PE/Cy7 anti-mouse CD11b (clone M01/70) | eBioscience | Cat: #25-0112-81 | FACS (1:200) |
| Antibody | Monoclonal Brilliant Violet 605-conjugated anti-CD11b (M1/70) | Thermo Fisher Scientific | Cat: #BDB563015 | FACS (1:400) |
| Antibody | Monoclonal FITC anti-mouse CD45 (clone 30-F11) | eBioscience | Cat: #11-0451-81 | FACS (1:200) |
| Antibody | Monoclonal BUV395 anti-mouse CD45 (clone 30-F11) | BD Biosciences | Cat: #564279 | FACS (1:400) |
| Antibody | Monoclonal PE/Cy7 anti-mouse CD45 (clone 30-F11) | eBioscience | Cat: #25-0451-82 | FACS (1:200) |
| Antibody | Monoclonal APC anti-mouse Ly-6C (clone HK1.4) | BioLegend | Cat: #128016 | FACS (1:150) |
| Antibody | Monoclonal APC/Cy7 anti-mouse Ly-6C (clone HK1.4) | BioLegend | Cat: #128025 | FACS (1:150) |
| Antibody | Monoclonal PE anti-mouse CSF1R (clone AFS98) | BioLegend | Cat: #135505 | FACS (1:100) |
| Antibody | Monoclonal Brilliant Violet 421-conjugated anti-Thy1.2 (clone 53-2.1) | BioLegend | Cat: #140327 | FACS (1:400) |
| Antibody | Monoclonal PEDazzle594-conjugated anti-CD19 (6D5) | BioLegend | Cat: #115553 | FACS (1:400) |
| Antibody | Monoclonal Brilliant Violet 711-conjugated anti-CD4 (RM4-5) | BioLegend | Cat: #100549 | FACS (1:200) |
| Antibody | Monoclonal Brilliant Violet 785-conjugated anti-CD8a (53-6.7) | BioLegend | Cat: #100749 | FACS (1:200) |
| Antibody | Monoclonal Brilliant Violet 650-conjugated anti-NK1.1 (PK136) | BioLegend | Cat: #108735 | FACS (1:400) |
| Antibody | Monoclonal Alexa Fluor 700-conjugated anti-CD3 (17A2) | BioLegend | Cat: #100215 | FACS (1:200) |
| Antibody | Monoclonal AF488-conjugated anti-FoxP3 (FJK-16s) | eBioscience | Cat: #53-5773-82 | FACS (1:200) |
| Antibody | Monoclonal PE-conjugated anti-Gata3 (TWAJ) | eBioscience | Cat: #12-9966-42 | FACS (1:100) |

*Continued on next page*

*Continued*

| Reagent type (species) or resource | Designation | Source or reference | Identifiers | Additional information |
|---|---|---|---|---|
| Antibody | Monoclonal anti-mouse CD16/32 antibody | eBioscience | Cat: #14-0161-82 | FACS (1:200) |
| Commercial assay or kit | Foxp3/Transcription Factor Staining Buffer Set | eBioscience (Thermo Fisher Scientific) | Cat. no.: 00-5523-00 | |
| Commercial assay or kit | RNeasy Plus Micro Kit | Qiagen | Cat. no./ID: 74034 | |
| Commercial assay or kit | Agilent RNA 6000 Pico Kit | Agilent | Part no.: 5067-1513 | |
| Commercial assay or kit | Ovation RNA-Seq System V2 Kit | NuGen | Part no.: 7102 | |
| Commercial assay or kit | Trio RNA-Seq Kit | NuGen | Part no.: 0506 | |
| Commercial assay or kit | Qubit dsDNA HS Assay Kit | Thermo Fisher Scientific | Cat no.: Q32851 | |
| Software, algorithm | Fiji (ImageJ) | *Schindelin et al., 2012* | RRID:SCR_002285 | |
| Software, algorithm | FastQC | Babraham Institute | RRID:SCR_011106 | |
| Software, algorithm | STAR (version 2.5.4b) | *Dobin et al., 2013* | | |
| Software, algorithm | HTSeq (version 0.9.0) | *Anders et al., 2015* | RRID:SCR_005514 | |
| Software, algorithm | DESeq2 (version 1.24.0) | *Love et al., 2014* | RRID:SCR_015687 | |
| Software, algorithm | Limma | *Ritchie et al., 2015* | RRID:SCR_010943 | |
| Software, algorithm | Metascape | *Zhou et al., 2019* | RRID:SCR_016620 | |
| Other | Zombie NIR (fixable viability dye) | BioLegend | Cat: #423105 | FACS 1:1000 |
| Other | DAPI | Sigma-Aldrich | Cat: #9542 | 1:1000 |
| Other | RLT+ | Qiagen | Cat: # 1053393 | |

## Mice

All mouse experiments were approved by UCSF Institutional Animal Care and Use Committee and conducted in accordance with the guidelines established by the Institutional Animal Care and Use Committee and Laboratory Animal Resource Center. All mice were between 8 and 14 weeks old when experiments were performed. Littermate controls were used for all experiments when feasible and all experiments were performed in male and female mice. WT (C57BL/ 6J) and Rag1 knockout (B6.129S7-*Rag1*tm1Mom/J; Stock no.: 002216) mice were purchased from The Jackson Laboratory. The following previously described strains were used and bred in house: *Csf1*fl/fl (*Harris et al., 2012*), *Avil*Cre (*Zurborg et al., 2011*), and *FoxP3*DTR (B6.129(Cg)-*Foxp3*tm3(DTR/GFP)Ayr/J) (*Kim et al., 2007*).

## Injury, injections, and behavioral analysis

Spared Nerve Injury (SNI) was performed by ligation and transection of the sural and superficial peroneal branches of the sciatic nerve, leaving the tibial nerve intact (*Shields and Eckert, 2003*). CSF1 (Life Technologies; PMC2044) was injected intrathecally at low dose (15 ng) or high dose (30 ng) in a total volume of 5 µl for three times over 3 days (24 hr between injections). Behavioral analysis was done 2 hr after injections; mice were euthanized for analysis about 4 hr after the last injection. All Von Frey behavioral experiments were performed during the light cycle as previously reported (*Guan et al., 2016*) in a blinded manner. Intraperitoneal injection of anti-CD4 (250 µg) (InVivoPlus; Bio X Cell) and Diphtheria toxin (30 ng/g) (Sigma-Aldrich) were all in a volume of 200 µl per injection. Anti-CD4 was given 1 day prior to the start of CSF1 injections, and on day 2 of the CSF1 injections. Diphtheria toxin was given 2 days (two subsequent injections) before the start of the CSF1 injections, and on day 2 of the CSF1 injections.

## Immunohistochemistry and analysis

Avertin-anesthetized mice were transcardially perfused with 1× phosphate-buffered saline (PBS) (~10 ml) followed by 4% (weight/volume) paraformaldehyde (PFA) diluted in PBS (~10 ml). Spinal cord tissue was dissected out and post-fixed in 4 % PFA for 4 hr and then transferred to a 30% sucrose solution overnight. Subsequently, spinal cords were sectioned coronally at 25 µm using a cryostat (Thermo Fisher Scientific). Spinal cord sections were incubated in a blocking solution consisting of 10% normal goat (Thermo Fisher Scientific) and 0.4% Triton (Sigma-Aldrich) diluted in 1× PBS. Primary antibodies included: rabbit anti-mouse Iba1 (WAKO, 1:2000); Alexa 647-coupled mouse anti-CD45 (BioLegend, 1:200); and hamster anti-CD3 (BD BioScience, 1:200). Antibodies were diluted in 10% normal goat with 0.4% Triton in PBS and incubated on a shaker overnight at 4°C. Secondary antibodies (Thermo Fisher Scientific, 1:1000) were diluted in 0.4% Triton in PBS and spinal cord sections were incubated on a shaker for 2 hr at room temperature. Spinal cord sections were mounted on coverslips with DAPI containing Fluoromount-G (Thermo Fisher Scientific). Slides were imaged on an LSM700 (Zeiss) confocal microscope using 63× objectives and z-stacks with a step size of 1 µm were collected. In Fiji (*Schindelin et al., 2012*) (ImageJ), maximum intensity images were generated and binary, thresholded images for morphology analysis were created. Subsequently, Scholl analysis (*Ferreira et al., 2014*) was done in Fiji (ImageJ) on microglia from the binary images with a step size of 2.5 µm.

## Fluorescence-activated cell sorting of microglia

To isolate microglia, we followed a previously described method (*Galatro et al., 2017*). Briefly, lumbar dorsal horn spinal cords were mechanically dissociated in isolation medium (HBSS, 15 mM HEPES, 0.6% glucose, 1 mM EDTA pH 8.0) using a glass tissue homogenizer (VWR). Next, the suspension was filtered through a 70 µm filter and then pelleted at 300 ×*g* for 10 min at 4°C. The pellet was resuspended in 22% Percoll (GE Healthcare) and centrifuged at 900×*g* for 20 min (acceleration set to 4 and deceleration set to 1). The myelin free pelleted cells were then incubated in blocking solution consisting of anti-mouse CD16/32 antibody (eBioscience) for 5 min on ice, followed for 30 min in a mix of PE or PE/Cy7-conjugated anti-mouse CD11b (eBioscience), FITC or PE/Cy7-conjugated anti-mouse CD45 (eBioscience), and APC or APC/Cy7-conjugated anti-mouse Ly-6C (BioLegend) in isolation medium that did not contain phenol red. For flowcytometric analysis of CSF1R expressed by microglia, PE-conjugated anti-mouse CSF1R (BioLegend) was added. The cell suspension was centrifuged at 300 ×*g* for 10 min at 4°C and the pellet was incubated with DAPI (Sigma-Aldrich) before sorting. Microglia were sorted on a BD FACS Aria III and gated on forward/side scatter, live cells by DAPI, and CD11b$^{high}$, CD45$^{low}$, and Ly-6C$^{neg}$. After sorting, cells were spun down at 500 ×*g*, 4°C for 10 min and the pellet was lysed with RLT+ (Qiagen).

## Isolation of spinal cord meningeal cells

Single-cell suspensions were prepared by digesting dissected spinal cord meninges with Liberase TM (0.208 WU/ml) and DNase I (40 ug/ml) in 1.0 ml cRPMI (RPMI supplemented with 1 10% (vol/vol) fetal bovine serum (FBS), 1% (vol/vol) Hepes, 1% (vol/vol) Sodium Pyruvate, 1% (vol/vol) penicillin-streptomycin) for 30–40 min at 37°C, 220 RPM. Digested samples were then passed over a 70 µm cell strainer and any remaining tissue pieces macerated with a plunger. Cell strainers were additionally

flushed with FACS wash buffer (FWB, PBS w/o $Mg^{2+}$ and $Ca^{2+}$ supplemented with 3% FBS and 0.05% NaN3). Single-cell suspensions were washed and resuspended in FWB.

## Flow cytometry of spinal cord meningeal cells

To exclude dead cells from the analysis, single-cell suspensions were stained with a fixable viability dye (Zombie NIR, BioLegend), followed by staining for surface antigens with a combination of the following fluorescence-conjugated mAbs: Brilliant Violet 421-conjugated anti-Thy1.2 (53-2.1) (BioLegend), PEDazzle594-conjugated anti-CD19 (6D5) (BioLegend), Brilliant Violet 605-conjugated anti-CD11b (M1/70) (Thermo Fisher Scientific), Brilliant Violet 711-conjugated anti-CD4 (RM4-5) (BioLegend), Brilliant Violet 785-conjugated anti-CD8a (53-6.7) (BioLegend), Brilliant Violet 650-conjugated anti-NK1.1 (PK136) (BioLegend), Alexa Fluor 700-conjugated anti-CD3 (17A2) (BioLegend), and BUV395-conjugated anti-CD45 (30-F11) (BD Biosciences). Cells were then fixed and permeabilized using the Foxp3/Transcription Factor Staining Buffer Set (eBioscience), followed by staining for intracellular antigens using the following mAbs (all from eBioscience): AF488-conjugated anti-FoxP3 (FJK-16s) and PE-conjugated anti-Gata3 (TWAJ). Samples were acquired on a Fortessa (BD Biosciences) and analyzed with FlowJo 10 software (BD Biosciences).

## RNA sequencing of microglia

RNA from RLT+ lysed microglia was isolated using the RNeasy Plus Micro Kit (Qiagen) and quality and concentration were assessed with the Agilent RNA 6000 Pico Kit on a Bioanalyzer (Agilent). For samples from male and female microglia collected from the saline or CSF1 injection data sets, cDNA and libraries were generated using the Ovation RNA-Seq System V2 Kit (NuGen). For samples from female Treg knockout or WT microglia collected from the CSF1 injection data set, cDNA and libraries were generated using the Trio RNA-Seq Kit (NuGen). Quality was determined with the Agilent High Sensitivity DNA Kit on a Bioanalyzer (Agilent) and concentrations were measured on Qubit (Thermo Fisher Scientific) with Qubit dsDNA HS Assay Kit (Thermo Fisher Scientific). Libraries were pooled and RNA sequencing was performed on an Illumina HiSeq 4000 with single-end 50 (SE50) sequencing. Between 40 and 60 million reads were sequenced per sample.

## RNA sequencing Analysis

Quality of reads was assessed using FastQC (http://www.bioinformatics.babraham.ac.uk/projects/fastqc) and all samples passed quality control. Subsequently, reads were aligned to mm10 (GRCm38; retrieved from Ensembl) using STAR (version 2.5.4b) (*Dobin et al., 2013*) without FilterMultimapNmax one so as to only keep reads that map one time to the reference genome. Uniquely mapped reads were counted using HTSeq (version 0.9.0) (*Anders et al., 2015*) and the DESeq2 package (version 1.24.0) (*Love et al., 2014*) in R was used to normalize the raw counts and perform differential gene expression analysis (using the apeglm method [*Zhu et al., 2019*] for effect size shrinkage). One CSF1-treated WT female sample was subsequently removed from the analysis as its counts significantly deviated from the rest. Specifically, its gene expression pattern resembled severe injury, potentially due to damage to the spinal cord during the mouse experimental procedures. Batch correction was done using the Limma package (*Ritchie et al., 2015*) in R. Volcano plot was generated using the EnhancedVolcano package (version 1.2.0), and the heatmap using ComplexHeatmap (*Gu et al., 2016*) in R. Metascape was used for GO analysis (*Zhou et al., 2019*). FPKM values were generated using Cufflinks (version 2.2.1) (*Trapnell et al., 2010*).

### Statistical analysis

For most statistical analyses, we used Graphpad Prism 8. Figure legends identify the specific statistical test used and additional details are provided in *Table 1*. RNA-sequencing data were analyzed in R as described in Materials and methods section.

## Acknowledgements

The authors are grateful to Michael Rosenblum and Ian Boothby advice on Treg depletion, and to the Basbaum and Molofsky labs for helpful comments on the manuscript. AIB is supported by R35 NS097306 and Open Philanthropy. AVM is supported by the Pew Charitable Trusts, NIMH (R01MH119349 and DP2MH116507), and the Burroughs Welcome Fund.

**Table 1.** Statistical reporting.

| Figure | N | Statistical test | Exact p-value | 95% confidence interval |
|---|---|---|---|---|
| *Figure 1b* | Male mice saline=3, male mice CSF1=6 | two-way ANOVA, repeated measures, Sidak's multiple comparison | Treatment=0.0009 | D1=34.46–75.46; D3=34.13–75.13; D5=8.754–49.75 |
| *Figure 1c* | Female mice saline=5, female mice CSF1=5 | Two-way ANOVA, repeated measures, Sidak's multiple comparison | Treatment=0.1890 | D1 = −30.41 to 10.58; D3=−28.58 to 12.41; D5=−30.13 to 10.86 |
| *Figure 1e* | 25 cells/group from 3 mice/condition | Unpaired t-test, two-tailed | Males<0.0001; females=0.0184 | Males=−309 to −195.1; females=16.78–174.6 |
| *Figure 1g* | Control males=10 mice, CSF1 males=9 mice, control females=10 mice, CSF1 females=10 mice | Ordinary one-way ANOVA, Tukey's multiple comparisons | Males<0.0001; females=0.0034; male vs. female CSF1=0.0002 | Males=−31.11 to −17.53; females=−15.81 to −2.591; male vs. female CSF1=4.976–18.56 |
| *Figure 1h* | Male saline=7, male CSF1=9, female saline=10, female CSF1=10 | Ordinary one-way ANOVA, Tukey's multiple comparisons | Males<0.0001; females=0.0677 | Males=−55.56 to −20.86; females=−30.01 to 77.82 |
| *Figure 1—figure supplement 1a.* | Female WT=6, female KO=5, male wt=9, male KO=7 | Unpaired students t-test for each sex | Females=0.2424; males< 0.0001 | Females=−11.33–39.37; males=62.83–114.3 |
| *Figure 1—figure supplement 1b.* | female WT = 6, female KO = 5, male WT = 3, male KO = 2 | N/A | N/A | N/A |
| *Figure 1—figure supplement 1c.* | 4 mice/group | Unpaired two-tailed t-test (each time point vs. baseline) | D1=0.2238; D2=0.1794 | D1=−0.2804 to 0.0804; D2=−0.2232 to 0.05217 |
| *Figure 2—figure supplement 1b.* | 4 mice/group | Two-way ANOVA | Interraction=0.2397, sex=0.3858, treatment=0.0501 | |
| *Figure 3b* | 5 mice/group | Unpaired t-test between males and females for each cell type | | |
| *Figure 3c* | Control=4 mice/sex, CSF1=5 mice/sex | One-way ANOVA, Tukey's multiple comparison test | Males=0.5422; females=0.0229 | Males=−595.9 to 215.9; females=−870.4 to −58.58 |
| *Figure 3h* | 5 mice/group | Unpaired two-tailed t-test | 0.22 | −5.987 to 22.16 |
| *Figure 3i* | WT=9 mice, FoxP3DTR=10 | Unpaired two-tailed t-test | 0.01 | −54.72 to −10.71 |
| *Figure 3j* | 4 mice/group | Unpaired two-tailed t-test | 0.04 | −65.47 to −1.634 |
| *Figure 3k* | 10 mice/group | Unpaired two-tailed t-test | 0.01 | −29.61 to −5.255 |
| *Figure 3—figure supplement 1b.* | Saline=4 mice/group, CSF1=5 mice/group | One-way ANOVA, Tukey's multiple comparison test | Males=0.4533; females=0.0111 | Males=−67049 to 21037; females=−100227 to −12141 |
| *Figure 3—figure supplement 1c.* | Saline=4 mice/group, CSF1=5 mice/group | One-way ANOVA, Tukey's multiple comparison test | Males=0.4797; females=0.0198 | Males=−10975 to 3602; females=−15820 to −1244 |
| *Figure 3—figure supplement 1e.* | No DT=2, DT=4 | N/A | N/A | N/A |

*Table 1 continued on next page*

*Table 1 continued*

| Figure | N | Statistical test | Exact p-value | 95% confidence interval |
|---|---|---|---|---|
| *Figure 3—figure supplement 1g.* | 5 mice/group | unpaired two tailed t-test | 0.2622 | –14.75 to 23.55 |
| *Figure 3—figure supplement 1i.* | 5 mice/group | Unpaired two-tailed t-test | 0 | –515 to –238.9 |

## Additional information

### Competing interests

Julia A Kuhn: Patent approved on use of CSF1 blockade to treat neuropathic pain (Publication Number WO/2016/057800).. Allan I Basbaum: Reviewing editor, eLife. The other authors declare that no competing interests exist.

### Funding

| Funder | Grant reference number | Author |
|---|---|---|
| National Institute of Neurological Disorders and Stroke | R35 NS097306 | Allan I Basbaum |
| Open Philathropy | | Allan I Basbaum |
| Pew Charitable Trusts | | Anna V Molofsky |
| National Institute of Mental Health | R01MH119349 | Anna V Molofsky |
| National Institute of Mental Health | DP2MH116507 | Anna V Molofsky |
| Burroughs Wellcome Fund | | Anna V Molofsky |

The funders had no role in study design, data collection and interpretation, or the decision to submit the work for publication.

### Author contributions

Julia A Kuhn, Conceptualization, Data curation, Formal analysis, Investigation, Methodology, Validation, Visualization, Writing – original draft; Ilia D Vainchtein, Conceptualization, Formal analysis, Investigation, Methodology, Validation, Visualization, Writing – original draft, Writing – review and editing; Joao Braz, Conceptualization, Data curation, Formal analysis, Investigation, Methodology, Validation, Visualization, Writing – original draft, Writing – review and editing; Katherine Hamel, Mollie Bernstein, Jorge Ortiz-Carpena, Data curation, Investigation, Methodology, Validation, Visualization; Veronica Craik, Investigation, Methodology; Madelene W Dahlgren, Data curation, Formal analysis, Investigation, Methodology, Validation, Visualization; Ari B Molofsky, Conceptualization, Formal analysis, Funding acquisition, Methodology, Project administration, Supervision, Writing – original draft, Writing – review and editing; Anna V Molofsky, Conceptualization, Data curation, Formal analysis, Funding acquisition, Investigation, Methodology, Project administration, Resources, Supervision, Validation, Visualization, Writing – original draft, Writing – review and editing; Allan I Basbaum, Conceptualization, Data curation, Formal analysis, Funding acquisition, Investigation, Project administration, Resources, Supervision, Validation, Visualization, Writing – original draft, Writing – review and editing

### Author ORCIDs

Ilia D Vainchtein  http://orcid.org/0000-0003-3190-8524
Mollie Bernstein  http://orcid.org/0000-0003-2327-5771
Anna V Molofsky  http://orcid.org/0000-0002-4709-2411
Allan I Basbaum  http://orcid.org/0000-0002-1710-6333

## Ethics

As noted in the description of the mice used in this study: "All mouse experiments were approved by UCSF Institutional Animal Care and Use Committee and conducted in accordance with the guidelines established by the Institutional Animal Care and Use Committee and Laboratory Animal Resource Center." Please note that this is a renewal that occurred during the course of the revision to the manuscript. APPROVAL NUMBER: AN183265-02D Approval Date: June 15, 2021 Expiration Date: February 26, 2022.

## Decision letter and Author response

Decision letter https://doi.org/10.7554/eLife.69056.sa1
Author response https://doi.org/10.7554/eLife.69056.sa2

# Additional files

## Supplementary files

• Supplementary file 1. Transcriptomic profiling of CSF1-induced genes in microglia in males and females (Excel file).

• Supplementary file 2. FPKM values for each sample from transcriptomic profiling of males and females after saline or CSF1 (Excel file).

• Supplementary file 3. Transcriptomic profiling of CSF1-induced genes in microglia from females with and without Treg depletion (Excel file).

• Transparent reporting form

## Data availability

RNA sequencing data are available through GEO accession #GSE 184801 All data generated or analysed during this study and required for conclusions to be drawn are included in the manuscript and supporting files. The upload can be identified at the following link: https://www.ncbi.nlm.nih.gov/geo/query/acc.cgi?acc=GSE184801.

The following dataset was generated:

| Author(s) | Year | Dataset title | Dataset URL | Database and Identifier |
|---|---|---|---|---|
| Vainchtein ID | 2021 | Regulatory T-cells inhibit microglia-induced pain hypersensitivity in female mice | https://www.ncbi.nlm.nih.gov/geo/query/acc.cgi?acc=GSE184801 | NCBI Gene Expression Omnibus, GSE184801 |

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
