## [Decision Letter]

**Acceptance summary:**

This manuscript is of broad relevance to the research areas of pain and neuroimmunology, and is of particular interest for its investigation of sex differences in pain sensation. The data support the major conclusions, and highlight a previously unappreciated role for specific types of immune cells residing in the spinal cord envelope and how they can influence the nervous system by acting on microglia within the spinal cord.

**Decision letter after peer review:**

Thank you for submitting your article "Regulatory T-cells inhibit microglia-induced pain hypersensitivity in female mice" for consideration by *eLife*. Your article has been reviewed by 3 peer reviewers, and the evaluation has been overseen by a Reviewing Editor and Kate Wassum as the Senior Editor. The following individuals involved in review of your submission have agreed to reveal their identity: Claire E Le Pichon (Reviewer #1); Long-Jun Wu (Reviewer #3).

The reviewers have discussed their reviews with one another, and the Reviewing Editor has drafted this to help you prepare a revised submission. We are pleased to say that all agree this is an interesting and exciting study that would be suitable for *eLife* after a few revisions. The consensus is that some additional information should be added to help clarify a few important details. Specifically, the reviewers ask for evidence relating to the role of Treg in microglia-mediated pain control in males and that you discuss how meningeal Treg remotely communicate with microglia in the parenchyma of spinal cord. Ideally, these answers would be given through the inclusion of new data, or where not possible, at least discussed. Please refer to the specific reviewers' comments for details. We look forward to reading your revised manuscript when it is ready.

*Reviewer #1:*

The authors start with the discovery of a sex difference in intrathecally injected (i.t.) CSF1-induced pain behavior. Females do not develop the hypersensitivity observed in males after this treatment. The authors then go after the mechanism that would explain this sex difference and focus on one facet of how this occurs (a sex-specific role of regulatory T cells).

In previous work (Guan et al. 2016), the same authors had investigated the mechanism(s) underlying pain sensitivity following nerve injury. They had shown that nerve injury causes upregulation of the cytokine Csf1 in sensory neurons and a microglial response in the spinal cord, and that deletion of this gene in these neurons prevents the microglial activation and the pain behavior. They hypothesized that CSF1 from sensory neurons is secreted at their spinal terminals where microglia expressing the CSF1 receptor (Csf1r) engage in signaling that ultimately results in pain hypersensitivity behavior. They also showed the sufficiency of CSF1 to produce at least transient pain behavior by i.t. injection of CSF1 protein in the absence of a nerve injury. However all this was performed only in male mice.

In the current study, the authors reproduce this data in males and discover an intriguing sex difference. In females, i.t. CSF1 (in the absence of nerve injury) does not cause microglial activation nor pain hypersensitivity as it does in males. An advance over their previous study is a more in-depth characterization of the nature of the microglial activation, using transcriptional profiling and morphological analysis. By both these readouts, the microglia are significantly less activated in females. The authors hypothesize this dampening of microglial activation in females results from regulation of microglia by other immune cells residing in the meninges. The authors investigate the lymphocyte composition in the spinal cord meninges after i.t. CSF1 and observe that regulatory T cells (Treg) increase to a greater extent in females than males, along with other changes in lymphocyte populations that are differentially altered by sex. They ask whether Tregs are responsible for dampening the microglial activation in females. Remarkably, Treg depletion by multiple methods results in females now resembling males, with an increased microglial activation and pain behavior following i.t. CSF1.

Overall, this data in this study convincingly supports the major conclusions that (1) the immune response to i.t. CSF1 differs between sexes, and (2) meningeal lymphocytes play important roles in regulating microglial function. The methods employed are well suited to test the authors' hypotheses, and the authors consistently use multiple methods to converge on a given result (e.g. microglial activation in Figures 1 and 2 or Treg depletion in Figure 3G-K) which makes the data very convincing overall. The microglial profiling by sex is of high interest and provides a resource for the community to mine and to dig deeper into how the microglial activation might alter von Frey reflexive behavior.

The follow-up investigation of how this microglial activation may be repressed in females is only partial. It is restricted just to the role of Tregs and only in female mice, which may provide an incomplete picture of the mechanisms involved. However, it is nevertheless convincing as an investigation of one aspect of meningeal lymphocyte-dependent regulation of microglial function that then impacts neural circuits and behavior.

In summary, this well-written and -presented study opens up many interesting questions and directions for future work in the pain field and more broadly for other neurological conditions investigating the influence of meningeal lymphocytes on parenchymal inflammation that may be differentially regulated by sex.

Comments for the authors:

One useful control would be to actually show independent data for the citation of reference 32 (Costigan et al) that "lymphocytes including T cells are only rarely detected in the spinal cord, even after nerve injury". Demonstrating this would help solidify the authors' novel finding that lymphocytes do not infiltrate the parenchyma, but rather signal from the meninges.

Another recommendation is an experiment that would clarify to what extent the i.t. CSF1 mirrors CSF1 upregulation in the context of SNI (and therefore validity for neuropathic pain). The authors could examine SNI-induced meningeal lymphocyte composition. One hypothesis for how i.t. CSF1 and SNI may differ mechanistically is the exact location of CSF1 release (parenchymal vs meningeal) and thus which CSF1R-expressing myeloid cells it is able to act on. In other words – does CSF1 originating from neurons actually cause meningeal lymphocyte proliferation? (In Figure S3 the authors show that myeloid cells increase in the meninges after i.t. CSF1.)

An acknowledgement of the complexity of the many differences shown in Figure 3B would be a welcome addition before focusing on the Tregs.

Along similar lines, is it possible the CSF-1 dependent increase in NK cells in females shown in Figure 3B might influence microglial function despite the pro-inflammatory role that is cited (line 189-190)?

A control experiment depleting Tregs in males is not included so it would be helpful to hear from the authors about why this may or may not have helped their study. From Figure 3B, Tregs do increase in males, although not to the same extent as in females. Similarly, would it be possible to induce Treg proliferation in males and test whether this now renders them female-like? A comment on the feasibility of these experiments would be appreciated, especially since the authors allude to conceptually similar strategies to potentially tune pain up or down (lines 257-259).

*Reviewer #2:*

The concept of crosstalk between spinal cord microglia and meningeal lymphocytes to control the sense of pain is intriguing and trendy in the field. The phenotypes they show in gender difference and Treg dependent pain control in females are stunning. This is a well-written and enjoyable manuscript for readers.

It will be reader's interest to know whether Tregs play a role in pain control only in females or also alleviate male's mechanical hypersensitivity at certain level. Although depletion of Treg in females recapitulates mechanical hypersensitivity induced by CSF1 in males, evidence suggests Treg may also play a role in males. For example, in figures 3 and 4, males show an increased Treg population after CSF1 injection compared to the saline control. In figure 3D, the PCA shows that microglial gene expression profiles in Treg depleted, CSF1 treated females are closer to "basal levels (saline treated)" in males, but are distant from "CSF1 treated" male microglia. The authors should include the male CSF1 treated, Treg depleted dataset to support the idea that Treg's effect is gender biased.

*Reviewer #3:*

In this study, Kuhn et al. investigated the sexual dimorphism in the contribution of microglia to chronic pain in mice. The study nicely extended their previous work on microglial CSF1 signaling in pain to sex dependence. Here, the authors started with an interesting phenomenon that CSF1 induced pain in female but not male mice. They further showed that CSF1 induced more immune activation in male than female mice. More interestingly, the study described CSF1-mediated cross talk between spinal cord microglia and lymphocytes from spinal cord meninges, and Treg can suppress the pain phenotype in female mice. Overall, the study is well designed and performed using multidisciplinary approaches, including behavioral test, FACs, RNAseq, and DTR mediated Treg ablation. Although it is relatively brief and lacks detailed mechanistic exploration, the results are exciting to understand an intriguing cellular mechanism of sex differences in neuropathic pain.

---

## [Author Response]

Reviewer #1:[…] Comments for the authors:One useful control would be to actually show independent data for the citation of reference 32 (Costigan et al) that "lymphocytes including T cells are only rarely detected in the spinal cord, even after nerve injury". Demonstrating this would help solidify the authors' novel finding that lymphocytes do not infiltrate the parenchyma, but rather signal from the meninges.

To validate the absence of substantial T cell infiltration upon sciatic nerve injury, we performed SNI in male and female mice. Seven days post injury, we only occasionally observed a CD3+ T cell in the lumbar region of the dorsal spinal cord. We added these data as Supplementary Figure 4 to the manuscript.

Another recommendation is an experiment that would clarify to what extent the i.t. CSF1 mirrors CSF1 upregulation in the context of SNI (and therefore validity for neuropathic pain). The authors could examine SNI-induced meningeal lymphocyte composition. One hypothesis for how i.t. CSF1 and SNI may differ mechanistically is the exact location of CSF1 release (parenchymal vs meningeal) and thus which CSF1R-expressing myeloid cells it is able to act on. In other words – does CSF1 originating from neurons actually cause meningeal lymphocyte proliferation? (In Figure S3 the authors show that myeloid cells increase in the meninges after i.t. CSF1.)

For a variety of reasons, we cannot provide information as to changes in meningeal Tregs after sciatic nerve injury. Even after intrathecal CSF1 we are detecting very few Tregs in the spinal cord meninges (Figures 3B,C). In contrast to intrathecal injections, sciatic nerve injury would induce a much more localized effect limited to the ipsilateral lumbar spinal cord making it very unlikely that we could detect a change in Tregs with a reasonable number of animals after SNI. For this reason, we respectfully request that this experiment not be required for the revision.

An acknowledgement of the complexity of the many differences shown in Figure 3B would be a welcome addition before focusing on the Tregs.

We modified the text to address this question.

Along similar lines, is it possible the CSF-1 dependent increase in NK cells in females shown in Figure 3B might influence microglial function despite the pro-inflammatory role that is cited (line 189-190)?

We added a short paragraph about NK cells to the discussion.

A control experiment depleting Tregs in males is not included so it would be helpful to hear from the authors about why this may or may not have helped their study.

To address this question, we repeated the experiment originally performed in females and now examined male mice. In contrast to female mice, ablation of Tregs using diphtheria toxin in Foxp3-DTR male mice did not alter mechanical withdrawal thresholds in response to intrathecal CSF1 (Suppl. Figure 3G). We conclude that Tregs do not modulate intrathecal CSF1-induced pain behavior in male mice.

From Figure 3B, Tregs do increase in males, although not to the same extent as in females. Similarly, would it be possible to induce Treg proliferation in males and test whether this now renders them female-like? A comment on the feasibility of these experiments would be appreciated, especially since the authors allude to conceptually similar strategies to potentially tune pain up or down (lines 257-259).

To induce Treg proliferation in spinal cord meninges of male mice, we injected i.p. and i.t. a combination of IL2/IL2RA antibodies, a well-established system to drive Treg proliferation. Unfortunately, injection of IL2/IL2RA alone resulted in a significant drop in mechanical thresholds compared to saline-treated mice. Intrathecal CSF1 after IL2/IL2RA treatment did not induce an additional drop in pain thresholds. Although this experiment might suggest that Tregs can inhibit CSF1-induced pain in male mice, we cannot draw this conclusion due to the lower pain thresholds in the IL2/IL2RA treated mice.

**Author response image 1. sa2fig1:** Inducing Treg proliferation with IL2/IL2RA alters mechanical withdrawal thresholds in male mice. (A:) Schematic showing the timeline to increase Treg proliferation in mice prior to intrathecal CSF1 injections. (B:) Change in mechanical withdrawal threshold in male control and IL2/IL2RA injected mice before and after 3 days of CSF1 i.t. All thresholds are normalized to baseline thresholds prior to IL2/IL2RA treatment.

Reviewer #2:[…] It will be reader's interest to know whether Tregs play a role in pain control only in females or also alleviate male's mechanical hypersensitivity at certain level. Although depletion of Treg in females recapitulates mechanical hypersensitivity induced by CSF1 in males, evidence suggests Treg may also play a role in males. For example, in figures 3 and 4, males show an increased Treg population after CSF1 injection compared to the saline control. In figure 3D, the PCA shows that microglial gene expression profiles in Treg depleted, CSF1 treated females are closer to "basal levels (saline treated)" in males, but are distant from "CSF1 treated" male microglia. The authors should include the male CSF1 treated, Treg depleted dataset to support the idea that Treg's effect is gender biased.

To better understand the role of Tregs in regulating CSF1-induced pain in male mice, we depleted Tregs in Foxp3-DTR male mice with diphtheria toxin, prior to intrathecal CSF1 injections. In contrast to female mice, we did not observe a difference in mechanical withdrawal thresholds between control and Treg depleted male mice (Suppl. Figure 3G), indicating that Treg depletion in males is not sufficient to alter pain thresholds in this model, even though we detected an increase in meningeal Tregs in response to CSF1. Based on this experiment, we believe that an additional microglia RNASeq gene expression analysis from Treg depleted male mice is unlikely to reveal key insights into sex specific pain signaling. For this reason, we hope that this experiment is not a requirement for the revision.